# Integrated Rogowski Coil Sensor for Press-Pack Insulated Gate Bipolar Transistor Chips

**DOI:** 10.3390/s20154080

**Published:** 2020-07-22

**Authors:** Chaoqun Jiao, Zuoming Zhang, Zhibin Zhao, Xiumin Zhang

**Affiliations:** 1School of Electrical Engineering, Beijing Jiaotong University, No.3 ShangYuanCun, Haidian District, Beijing 100044, China; xmzhang@bjtu.edu.cn; 2Rizhao Power Supply Company of State Grid Shandong Electric Power Company, Yantai Road 68, Donggang District, Rizhao, Shandong 276800, China; 18126182@bjtu.edu.cn; 3State Key Laboratory of Alternate Electrical Power System with Renewable Energy Sources, North China Electric Power University, No.2, Beinong Road, Changping District, Beijing 102206, China; zhibinzhao@ncepu.edu.cn

**Keywords:** Press-Pack IGBT (PPI), Chip reliability, PCB Rogowski Coil, Current measurement

## Abstract

Recently, the press-pack insulated gate bipolar transistor (IGBT) has usually been used in direct current (DC) transmission. The press-pack IGBT (PPI) adopts a parallel layout of boss chips, and the currents of each chip will be uneven in the process of turning on and off, which will affect the reliability of the device. To measure the currents of each chip, based on the analysis of the principle and equivalent model of the Rogowski coil, this paper puts forward the design scheme and design index of multi-layer printed circuit board (PCB) Rogowski coil with good high-frequency performance, strong anti-interference ability and sufficient sensitivity. With the simulation analysis of Altium Designer and ANSYS softwares, a 1 mm thick, 76-turn integrated four-layer PCB Rogowski coil is designed. Then, adding a composite integrator, an integrated Rogowski coil sensor for measurement of PPI chips currents is designed. The Pspice simulation and the experiment results show that the sensor is fully satisfied with the chip current measurement.

## 1. Introduction

Flexible direct current (DC) transmission has been widely developed with its more flexible and reliable transmission mode. In order to maintain the safe operation of the system, more reliable requirements are put forward for the controllable rectifiers, usually an insulated gate bipolar transistor (IGBT), which constitute the flexible converter. Compared with welded IGBT, the press-pack IGBT (PPI) has a longer service life, a unique short-circuit failure mode, and high current-carrying capacity. Generally, multiple chips of PPI are connected in parallel to increase the PPI current level. However, due to the layout of the boss chips, it will inevitably make the current of each chip unequal and the internal chip current overshoot during the turn-on and turn-off, This will affect the reliability of the device. For reliability of the PPI, it needs to measure the currents of each chips in the PPI [1,2,3,4,5,6,7,8,9,10].

A Rogowski coil current sensor is widely used to monitor the transient current in power electronic applications without external magnetic field interference. The Rogowski coil can be divided into a hard Rogowski coil, flexible Rogowski coil and PCB Rogowski coil. The hard Rogowski coil adopts a rigid framework, which is not easy to deform, with dense winding, high accuracy and poor adaptability. The flexible Rogowski coil adopts a flexible framework, which can be deformed, is easy to measure, and has sparse winding, but the winding accuracy is not as good as that of the hard coil. A PCB Rogowski coil printed on a PCB is not easy to deform, and its winding accuracy is greatly improved, but the space is limited, the cross-sectional area is small, and the number of turns is small, resulting in a small mutual inductance. Because they are hollow coils, they can be directly nested on the conductor under test, and the alternating current (AC) can be measured without contact.

As early as 1989 and 1991, Li Haiyan designed a Rogowski coil for measuring steep pulses. The coil adopts self integration and the sensitivity is 1 mV/A [11,12]. In 2008, Wei Bing et al. undertook a detailed analysis of the frequency response of the passive integrator, and proposed the design of the coaxial integrator with through-hole ceramic chip capacitor and multiple resistors in series to reduce the stray parameters, achieving a good effect, with a frequency band of 0.18–17.3 MHz [13]. In 2011 and 2013, D. Gerber, T. Guillod designed a PCB Rogowski coil with a coil bandwidth of more than 28 MHz and a time delay of 11 ns for over-current detection of a 4.5 kV PPI gate drive circuit. In this paper, the equivalent model of Rogowski coil and the prediction of parameters are established, and the accuracy of prediction is verified by measurement [14,15]. In 2013, Dr. Xiang Minjiang designed a passive and active composite integrated circuit with a bandwidth of 10 Hz~1 MHz for the acquisition of traveling wave signals in intelligent substations [16]. In 2016, Yadong Liu designed a PCB Rogowski coil for overhead transmission [17]. In 2016, Tao Tao designed a PCB Rogowski coil for the analysis of the anti-interference property [18]. In 2016, M. Tsukuda designed a micro PCB Rogowski coil for current monitoring and protection of a high-voltage power module. In 2017, they proposed a new idea for the clamp type PCB sensor, which is used for the current measurement of the intelligent power module and power electronic converter [19,20,21]. In 2016, Yang Jun et al. designed an active and flexible Rogowski coil, which is used for the measurement and collection of lightning current. The bandwidth is from hundreds Hz to tens of MHz, solving the problem of low sensitivity of passive RC integration [22]. From 2018 to the present, many researcher studied the multilayer PCB Rogowski coil for the measurement of precise impulse current and lightning current [23,24,25,26,27]. However, these coils are circular and can only measure precise impulse current and lightning current. The mutual inductance values of a square coil are larger than a circular coil with the same number of turns and cross-section area, which shows that square mutual inductance has more advantages. Moreover, to measure the multi-chip currents in the IGBT, a 4-coil integrated PCB Rogowski coil sensor is designed in this paper, but the coils in [23,24,25,26,27] are a single coil.

Due to the characteristics of large voltage and high current, the PPI is suitable for the converter of power systems. The reliability of the PPI is of great significance for the safe operation of the system. In this paper, an integrated Rogowski coil current transformer based on PCB is designed to monitor the breaking current of PPI in real time. Because the sensor is embedded, it can not only detect the current without changing the structure of the tested system, but also ensure the safe and reliable operation of the power system. As a kind of current sensor, it can also provide reference for the analysis of PPI characteristics, such as the current-sharing characteristics of PPI.

## 2. Design and Simulation of PCB Rogowski Coil Sensor

The working principle of the Rogowski coil and establishment of Rogowski coil equivalent model has been developed in [28,29,30,31]. According the characters of the currents in the IGBT, the design index of the sensor is the sensitivity (above 40 mV/A), bandwidth (not below 2 MHz) and measurement capability (0–100 A). 

### 2.1. The Design of the Single PCB Rogowski Coil 

Figure 1 shows the 4.5 kV/1.2 kA crimped IGBT structure of westcode, UK. It is composed of 11 IGBT chips and 5 anti parallel diodes with gaps. Each chip is 9.4 mm × 9.4 mm × 9.4 mm. 

According to the structure of a PPI boss, two kinds of coils of circular and square structure are designed as shown in Figure 2. Under certain space conditions, the circle is more in line with the trend of the magnetic field, while the square is more in line with the IGBT boss structure, and the space utilization rate is large. The square and circular coils with the same number of turns and cross-sectional area are simulated in ANSYS. The measured mutual inductance values are 2.7112 and 2.1634, respectively. The square mutual inductance has more advantages. Therefore, this paper finally adopts a 4-layer square plate structure with keel and return line, which can remove the unnecessary magnetic flux path of the wire turn and return line, reduce the influence of external current, effectively reduce the external signal interference, and make PCB technology reach the maximum limit. The structure of each layer of the Rogowski coil with square PCB is shown in Figure 3.

The PCB Rogowski coil is processed by the PCB manufacturer, and the finished product is shown in Figure 4. The pad part is used for welding terminal resistance, and the through hole is used for connecting the external circuit.

### 2.2. Design of the Integrated PCB Rogowski Coil

Multiple single coils can be used to measure the multi chip current, but the operation is tedious and the error is easily increased due to improper human placement and other factors. In order to solve the above problems, this paper integrates multiple coils into a PCB, which not only can realize the measurement of a multi-chip current, but also has the advantages of simple operation, high integration and small error. As shown in Figure 5, according to the size of the PPI module, a three-dimensional structure diagram of the PPI is established. Two excitation sources, source and sink, are set at the top and bottom of the notch boss (single IGBT chip). Four special bosses are selected for joint simulation to calculate the mutual inductance between the coil and 11 bosses. The arrangement is shown in Table 1.

It can be seen from the table that there is mutual inductance between the coil and the surrounding boss, and the closer the distance is, the greater the mutual inductance value is, but it is far less than the mutual inductance between the measured conductor and the coil, and the error is very small, which also shows that the PCB coil designed in this paper has certain anti-interference ability. The PCB board designed by Altium Designer(AD) and the actual processing drawing are shown in Figure 6.

### 2.3. Impedance Analysis of Rogowski Coil

The impedance analysis of PCB Rogowski coil can extract the corresponding electromagnetic parameters and analyze the amplitude frequency phase frequency characteristics of a PCB Rogowski coil. Two impedance analysis methods, the simulation method and the experimental method, are used to analyze and compare the coil, as shown in Figure 7.

After verifying the impedance of a single coil, the scanning analysis of four coils in the frequency band of 40 Hz–100 MHz is carried out in this paper. The frequency characteristic curve of the coil impedance is shown in Figure 8.

It can be seen from the frequency curve that the impedance and phase frequency of the four coils of the PCB integrated board designed in this paper are basically the same in the frequency band of 40 Hz–100 MHz, and there is no resonance point. To some extent, it avoids the parameter difference caused by the uneven number of hand winding turns, and proves the consistency of the parameters of the integrated coil. Combined with the experiment and simulation analysis, the final PCB coil parameters are shown in Table 2.

## 3. Overall Circuit Design and Simulation Based on Compound Integrator

Passive integration is suitable for measuring high-frequency current, and its working characteristics are relatively poor at low-frequency. Active integration is just the opposite, which is suitable for measuring low-frequency current, and its high-frequency characteristics are relatively poor. In order to realize the accurate measurement of the chip current of the PPI module, the integrator should be designed to realize the reduction of PCB Rogowski coil differential signal in a wide frequency band. Based on the above analysis, this paper selects a scheme of composite integrator, which combines active integration and passive integration as shown in Figure 9. In the low-frequency segment, active integration is responsible for integration, while in the high-frequency segment, unit gain has no effect on the signal; in the high-frequency segment, passive integration is responsible for integration, while in the low-frequency segment, unit gain has no effect on the signal.

When the measured current *i* is induced to the Rogowski coil, the Rogowski coil outputs a current differential signal, which is transmitted to the passive integrator through the coaxial cable, integrating the high-frequency part of the signal, integrating the low-frequency part of the signal through the in-phase active integrator, and filtering the low-frequency noise and DC drift generated by the operational amplifier through the high-pass filter The filter further filters the low-frequency noise of the signal, amplifies the signal through the buffer amplifier, and finally outputs the signal to the oscilloscope for display.

### 3.1. Simulation Analysis of Alternating Current (AC) Scanning of PCB Rogowski Coil Sensor

The overall circuit diagram of PCB Rogowski coil sensor is established in Pspice, and the amplitude frequency phase frequency characteristic diagram is obtained through AC frequency scanning analysis (see Figure 10 below). It can be seen from the phase frequency characteristic diagram of the sensor that the gain of the integrator is kept at about -28 dB in the frequency band of 100 Hz–5 MHz and the sensitivity is about 40 mV/A. From the phase frequency characteristic diagram, it can be seen that there is a constant 0° phase in the frequency band of 200hz-3mhz. It is worth noting that the blue curve data1 in the figure has high-frequency reflection in the high-frequency section. By adjusting the position of the passive integral resistance, the resistance *R*_P_ is moved to the coaxial cable end to solve this problem. After improvement, the data2 curve is obtained.

### 3.2. Transient Simulation Analysis of PCB Rogowski Coil Sensor

After analyzing the frequency domain of the coil sensor, the sensor is simulated in the time domain to verify the feasibility of the integrator circuit design. The current source in Pspice is changed into different signal sources, such as sine current source (Isin) and pulse signal current source (Pulse). Different frequency and amplitude waveforms can be obtained by setting different parameters.

(1) Simulation of sinusoidal current

The sinusoidal signal current source has three parameters: bias value (IOFF), peak amplitude (IAMPL) and frequency (FREQ). By setting the above three parameters, four special frequencies (100 Hz, 500 Hz, 5 MHz and 20 MHz) are selected, and the current amplitude is 1 A. Select the general settings under time domain (transient), set the running time and step size according to 1000 interval points of 6 cycles, and select 2 cycle waveforms to obtain the following waveforms, as shown in Figure 11.

It can be seen from the sine wave in Figure 11 that when the frequency is 100 Hz and the current input 1 A, the output is 0.04 V in amplitude, and the phase ratio is slightly ahead; when the frequency range is 500 Hz–5 MHz, the input and output are consistent, and the sensitivity is kept at 40 mV/A; when the frequency is 20 MHz, the output voltage is small, and the phase is shifted backward, the overall output is shifted, and the sensitivity is reduced.

(2) Simulation of square wave current 

By setting different parameters of pulse signal current source (Pulse), the square wave current signal with the same amplitude (1A) and different frequency is obtained as the measured current of the sensor, and the waveform is shown in Figure 12.

It can be seen from the square wave signal that the output has a large distortion at 100 Hz, the phase is slightly ahead of the input value, and the output phase is slightly behind the input value at 20 MHz. In the simulation of 5 kHz–5 MHz frequency, the input and output are close to coincidence. When the input peak value is 1 A, the output is 0.04 V and the sensitivity is kept at 40 mV /A, which can reflect the input current well.

(3) Simulation of triangle wave current

In addition to square wave signal, triangle wave current signal with the same amplitude (1 A) and different frequency can be generated as the measured current of the sensor by setting the parameters of pulse signal current source (Pulse). The waveform is shown in Figure 13.

It can be seen from the triangle wave signal that the phase is slightly ahead at 100 Hz, the output amplitude is distorted, 0.005 V lower than the ideal value, and the error is 12.5%; at 10 MHz, the output phase is slightly behind, 0.003 V lower than the ideal value, and the error is 7.5%. In the simulation of 5000 Hz–5 MHz frequency, the input and output are close to coincidence, which can reflect the input current very well and keep the sensitivity at 40 mV/A.

According to the simulation analysis of the three waveforms, in the low frequency (100 Hz), the square wave signal distortion is the most serious, and in the high frequency (10 MHz–20 MHz), the performance of the three waveforms is similar. In accordance with the simulation analysis of the amplitude frequency phase frequency of the coil, the sensor has a good measurement ability in the wide frequency of 500 Hz–5 MHz, and can reflect the measured current well. The results show that the designed PCB Rogowski coil sensor meets the requirements of the switching current frequency of the PPI chip.

In order to further verify the measurement ability of the Rogowski coil sensor for large current, two current levels of 300 A and 400 A are simulated by Pspice with the frequency of 1 MHz. The waveform is shown in Figure 14 below.

It can be seen from the above figure that the Rogowski coil sensor can respond well to the measured current in the range of 1 A–300 A. When the current increases to 400 A, the current output is distorted. It is not difficult to find that the ability of measuring current depends not only on the coil integrator itself, but also on the power supply and coil sensitivity of the operational amplifier. Because the sensitivity of the designed sensor is 40 mV/A and the supply voltage of op amp is ±15 V, the maximum current is limited to 375 A. So when the current reaches 400 A, the waveform is distorted. To sum up, the PCB Rogowski coil sensor designed in this paper can realize the current measurement at 1 A–300 A, 500 Hz–5 MHz, with a sensitivity of 40 mV/A, which meets the measurement requirements of the breaking current of the PPI chip. Compared with the commercial sensor, it has higher sensitivity and wider measurement range.

## 4. Verification of PCB Rogowski Coil Sensor

### 4.1. Experiment of Surge Catastrophe Signal of Rogowski Coil Sensor

As a standard test instrument, the lightning surge generator is a kind of pulse high-voltage generator which can simulate the time parameters of lightning waves. It can be used to evaluate the anti-interference ability of electrical equipment and electronic equipment. Based on the switching characteristics of the PCB Rogowski coil and compression IGBT designed in this paper, the lightning wave generated by a lightning surge generator is a good choice. Based on the laboratory conditions, lsg-506b of Shanghai Lingshi Electronics Co., Ltd. is selected in this paper, which can generate 10/70 µs open circuit voltage waveform, with the peak value up to 6 kV [32]. The main parameters are shown in Table 3, and the definition of output waveform is shown in Figure 15.

We select the waveform measured by the Rogowski current waveform transformer (CWT) UM-1 of the PEM company as the reference waveform, put the commercial Rogowski coil and self-made PCB Rogowski coil through the wire, and output them to the oscilloscope. It should be noted that in order to reduce the error, the conductor should be in the center of the Rogowski coil and perpendicular to the plane of the Rogowski coil. In order to verify the differential characteristics of the PCB coil, the output of the coil needs to be directly connected to the oscilloscope through coaxial cable.

We check the wiring, connecting the power supply, and constantly adjust the amplitude of the output voltage of the lightning surge generator; the current waveform can be observed through the oscilloscope. 

When the voltage is added to 300 V, the primary waveform is triggered. After data processing by MATLAB, the output waveform is as shown in Figure 16. The red curve represents the differential voltage waveform of the PCB Rogowski coil, and the blue represents the PEM Rogowski coil. It can be seen from the figure that the current waveform rises rapidly from 0 to the highest value and then drops, the output of the self-made coil rises sharply to the highest value and then decreases, then decreases to 0, then increases in reverse, and finally increases to 0. The differential characteristics of PCB Rogowski coil are verified.

Import the data of oscilloscope into MATLAB, process the data integration, and obtain the measured curve of single PCB coil as shown in Figure 17.

As can be seen from the Figure 18, compared with commercial sensors, the curve processed by integration in this paper can restore the measured current well, and the waveforms of the two are basically the same.

The trigger interval time is set to 20 s. For each integrated PCB Rogowski coil, the numbers are coil1, coil2, coil3 and coil4 (as shown in Figure 19 below). The four coils trigger a surge signal under the voltage of 600 V. After the integration processing, the curve as shown in Figure 19 above is obtained. It can be seen that the measurement results of the four integrated PCB coils are consistent. Through the lightning surge experiment, the advantages of PCB processing are demonstrated. The PCB ensures the consistency of each coil parameter, and further verifies the feasibility and performance of the integrated PCB Rogowski coil.

### 4.2. The Measurement Experiment of the Transient Current of the Crimp IGBT Chip

At present, the commonly used dynamic test platform for PPI is double the pulse test circuit (as shown in Figure 20 below), which conforms to IEC60747-9 test standard [33]. The circuit structure is simple, and only one pulse drive signal is needed in the test to realize the disconnection of IGBT, and it is more convenient to study the characteristics of IGBT. In this paper, a dual pulse dynamic test platform is built based on the laboratory conditions, and the PCB coil designed by ourselves is used to measure the transient current of the internal chip in order to further verify the performance of the designed PCB Rogowski coil. 

We turn on the power, turning on the switch k1, charge the high-voltage capacitor to 300 V, then turn off the switch k1 with the controller, trigger the optical pulse generator at the same time, and then driving the PPI, connect the PCB Rogowski coil into the oscilloscope through the coaxial cable in advance, thereby observing the current waveform of the IGBT collector. It should be noted that in order to further ensure the safety of the experiment, the power ground wire, high-voltage probe ground wire and the V_CE_E end of IGBT are all connected with the laboratory ground wire in this experiment.

The main research of the double pulse experiment is the waveform of the first turn-on and the second turn-on of the chip. As shown in Figure 21, when the first tube of the double pulse IGBT is turned off, the waveform measured by PCB Rogowski coil suddenly drops and then rises to 0; at the second turn-on, the measured voltage first rises to the highest point, then rapidly drops to 0, then drops in reverse to the lowest point, and finally returns to 0. This is a good verification of the differential characteristics of the designed PCB Rogowski coil.

After integrating the waveform data, the current off and on waveforms are obtained as shown in Figure 22 and Figure 23. It can be seen that the waveforms are basically consistent with the current waveforms measured by commercial CWT Rogowski coil. In the turn-off waveform, analyzing the current drop time, i.e., the time from 0.9 times collector current to 0.1 times collector current, it can be calculated that the current drop time is about 0.3 μs and the bandwidth is about 3 MHz, which is in the frequency band of the composite integrator designed in this paper. During the opening period, due to the influence of parasitic parameters of anti-parallel diodes and inductors, a overshoot of up to 65 A occurs during the opening process. It should be noted that if the overshoot is higher than the specified value of the chip, the reliability of the device will be affected. Therefore, the PCB Rogowski coil designed in this paper can not only monitor the transient breaking current of the crimped IGBT, but also provide a reference for the reliability research of the PPI module.

## 5. Conclusions

In view of the possible uneven current and internal chip current overshoot in the process of switching on and off of each chip of the press-pack IGBT, in this paper a current measurement sensor based on a multilayer PCB Rogowski coil is introduced. On the basis of eliminating large measurement error caused by uneven winding of the traditional Rogowski coil, the sensitivity and anti-interference ability of output voltage are further improved. The integrated PCB Rogowski coil is only 1 mm thick, which can measure the multi-chip current, and accurately reflect the impulse current waveform under lightning surge and the breaking current waveform of the press-pack IGBT. The sensitivity of the sensor is 40 mV/A, the current measurement capacity is 0–375 A, and the high-frequency bandwidth is about 3 MHz, which can meet the design requirements and current measurement of the IGBT chip. Compared with the standard commercial sensor, the device has lower cost, higher sensitivity, wider measurement capability and easier integration, which provides an important reference for the reliability research of the device.

## Figures and Tables

**Figure 1 sensors-20-04080-f001:**
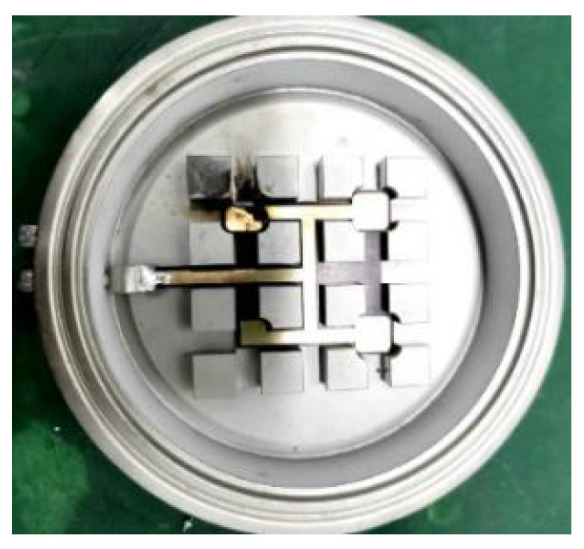
Outline structure of press-pack insulated gate bipolar transistor (PPI).

**Figure 2 sensors-20-04080-f002:**
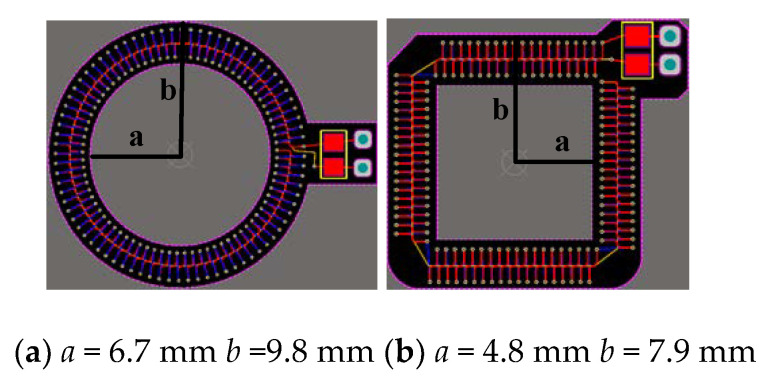
Dimensional drawings of Rogowski coils with different structures.

**Figure 3 sensors-20-04080-f003:**
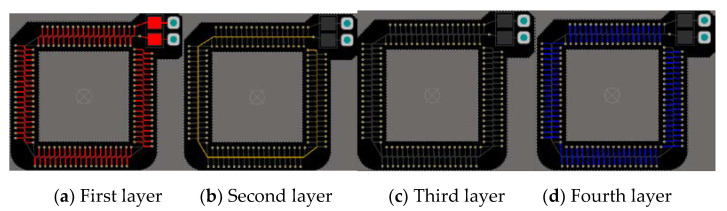
Fishbone-type four-layer board design pattern with return line.

**Figure 4 sensors-20-04080-f004:**
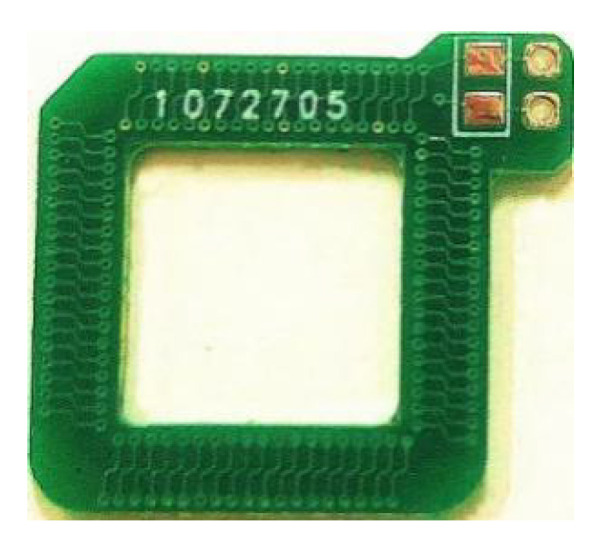
PCB Rogowski Coil physical map.

**Figure 5 sensors-20-04080-f005:**
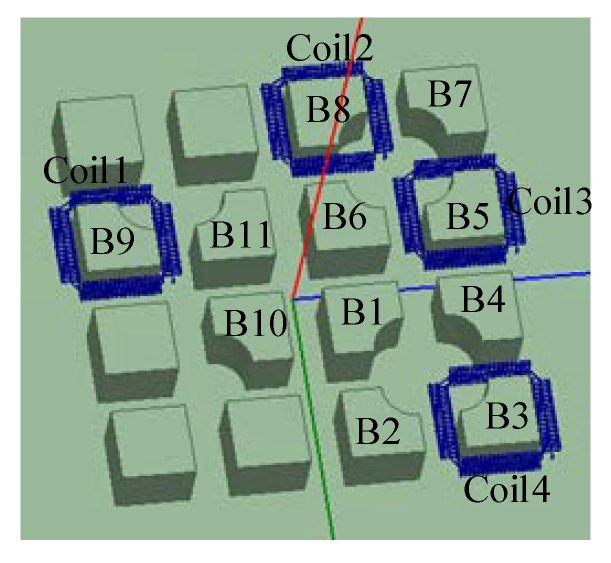
Joint simulation model.

**Figure 6 sensors-20-04080-f006:**
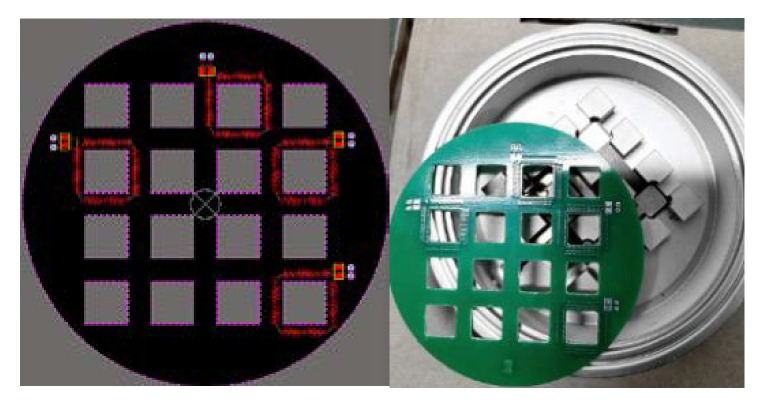
Integrated PCB design and physical drawing.

**Figure 7 sensors-20-04080-f007:**
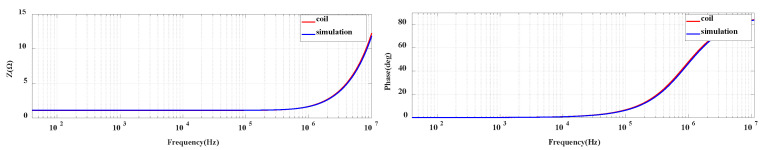
Comparison of simulation and experimental impedance analysis of coil head.

**Figure 8 sensors-20-04080-f008:**
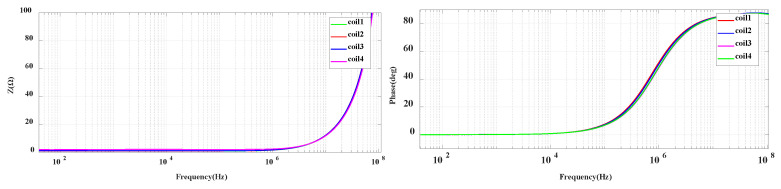
Impedance analysis of integrated PCB coils.

**Figure 9 sensors-20-04080-f009:**
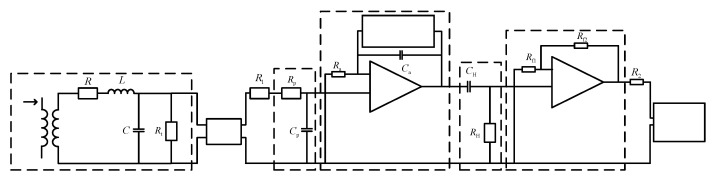
PCB Rogowski coil sensor overall design circuit diagram.

**Figure 10 sensors-20-04080-f010:**
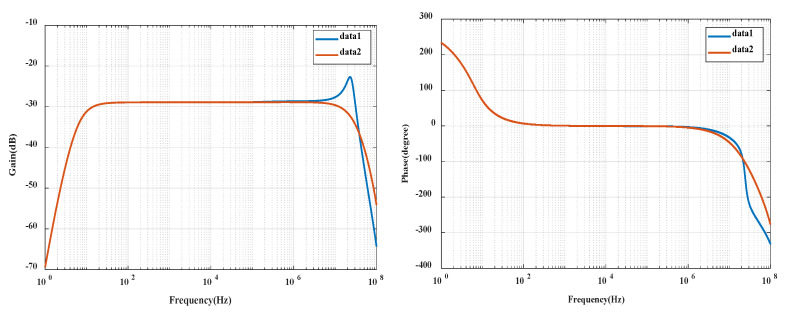
Coil sensor overall amplitude frequency phase frequency characteristic diagram.

**Figure 11 sensors-20-04080-f011:**
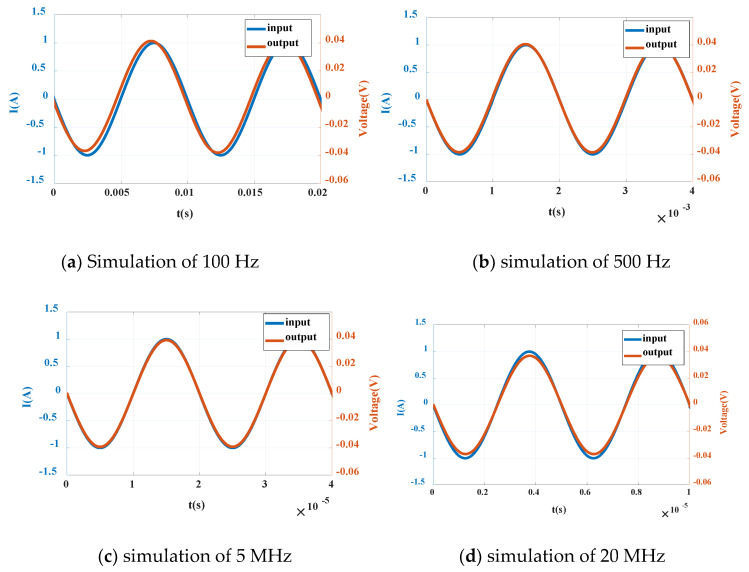
Sine wave simulation diagrams at different frequencies.

**Figure 12 sensors-20-04080-f012:**
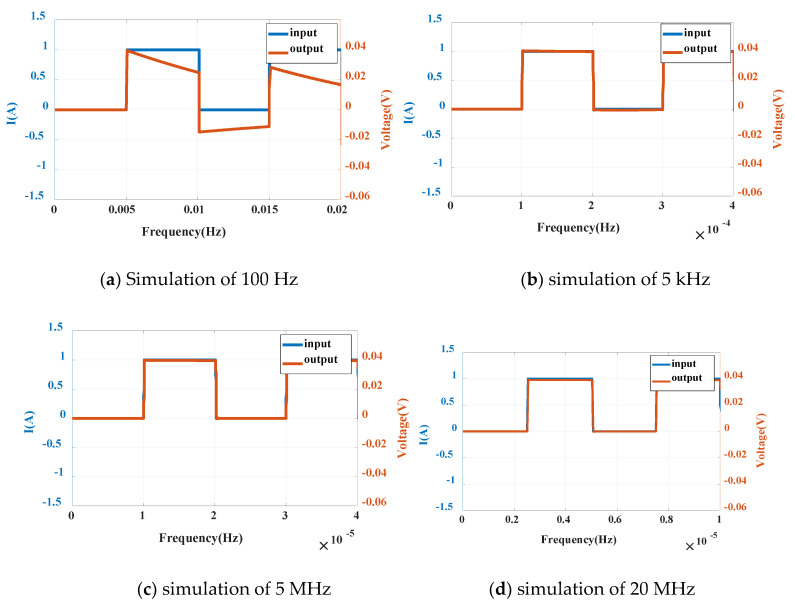
Simulation of square wave at different frequencies

**Figure 13 sensors-20-04080-f013:**
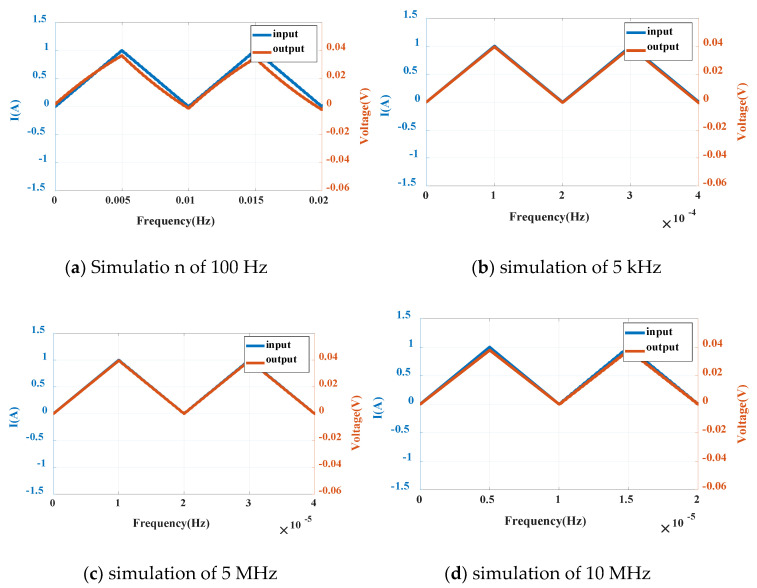
Triangular wave simulation diagrams at different frequencies

**Figure 14 sensors-20-04080-f014:**
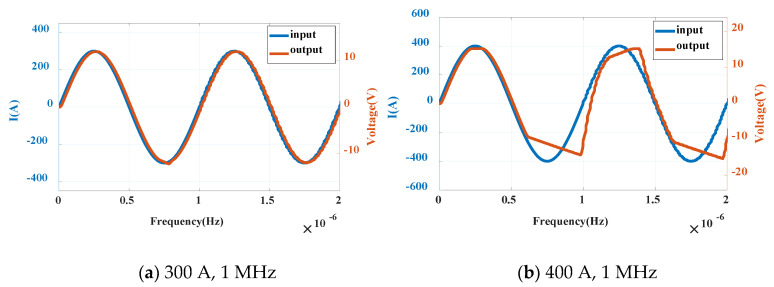
Sine wave simulation diagrams under different currents

**Figure 15 sensors-20-04080-f015:**
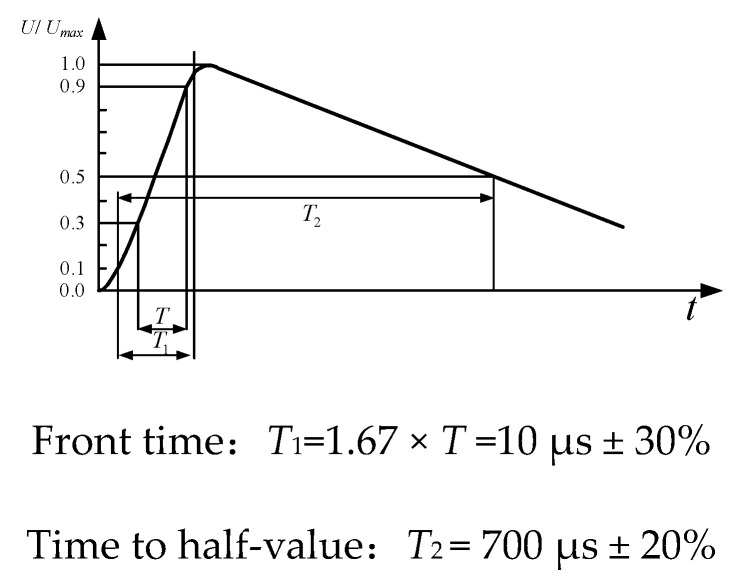
Output waveform CCITT of LSG-506B.

**Figure 16 sensors-20-04080-f016:**
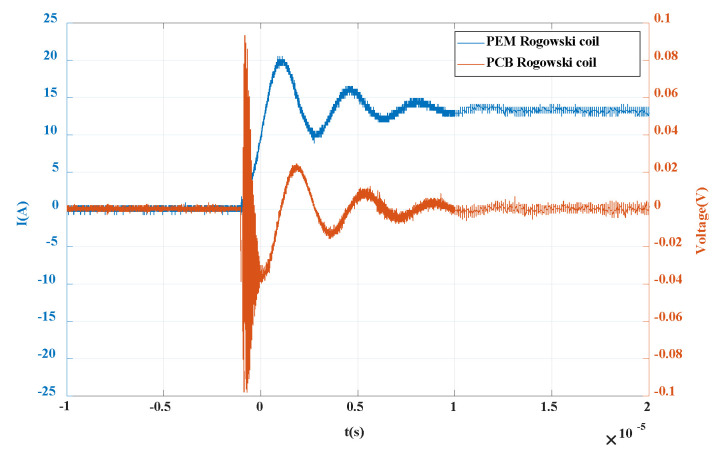
Differential verification waveform of PCB Rogowski coil.

**Figure 17 sensors-20-04080-f017:**
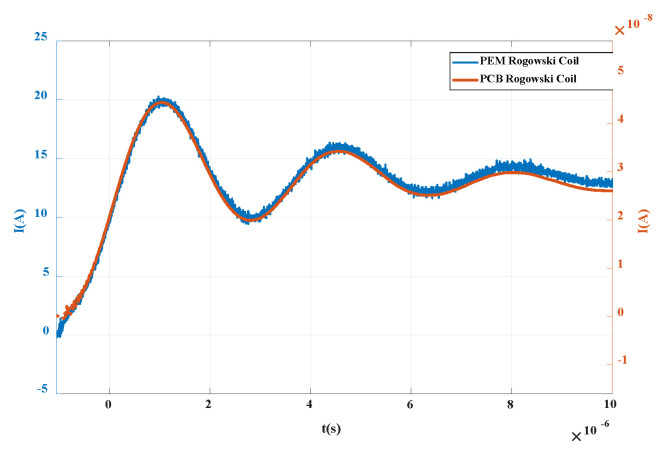
Integral waveform of PCB Rogowski coil.

**Figure 18 sensors-20-04080-f018:**
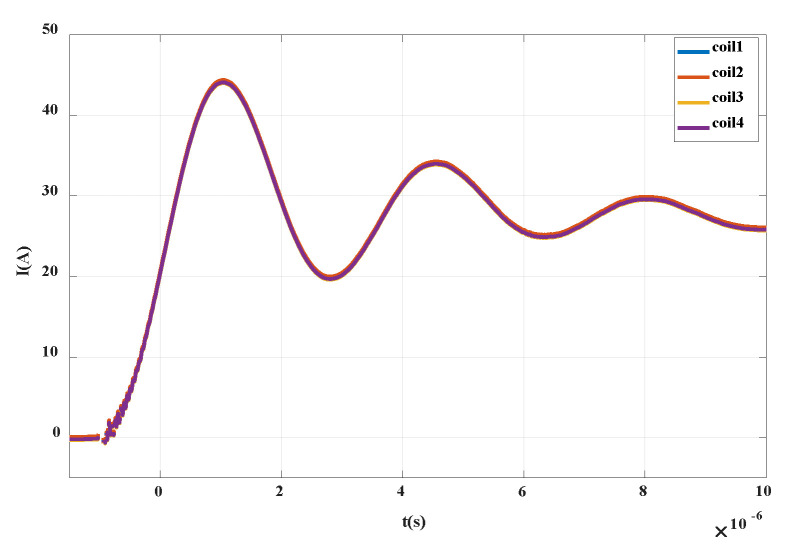
Integrated PCB Rogowski coil integral waveform.

**Figure 19 sensors-20-04080-f019:**
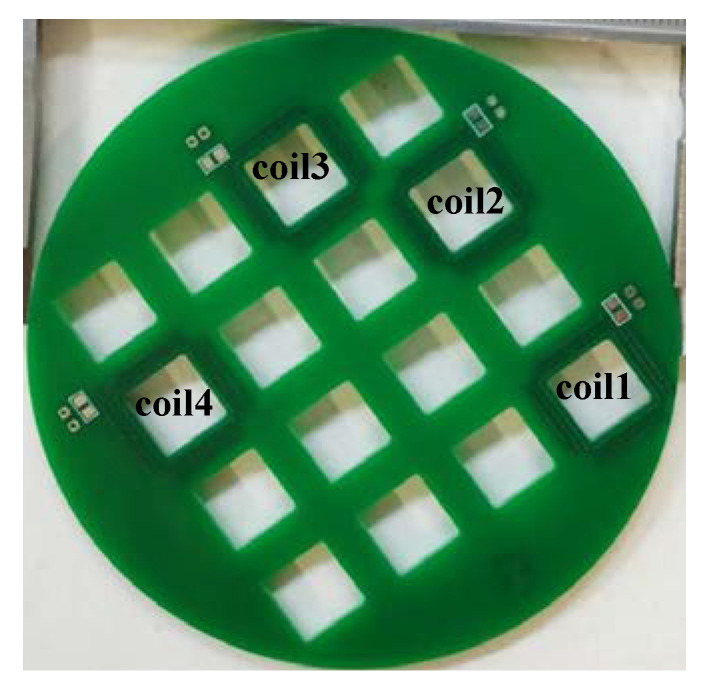
Physical number of integrated PCB Rogowski coil.

**Figure 20 sensors-20-04080-f020:**
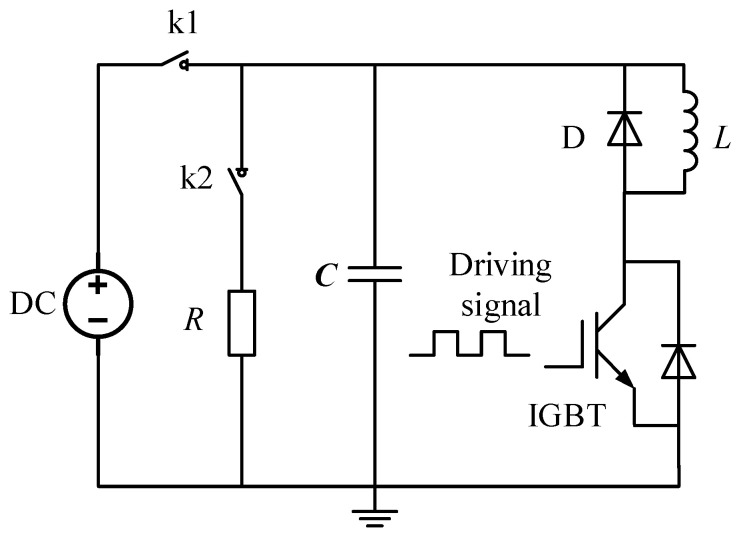
Dual pulse test platform schematic.

**Figure 21 sensors-20-04080-f021:**
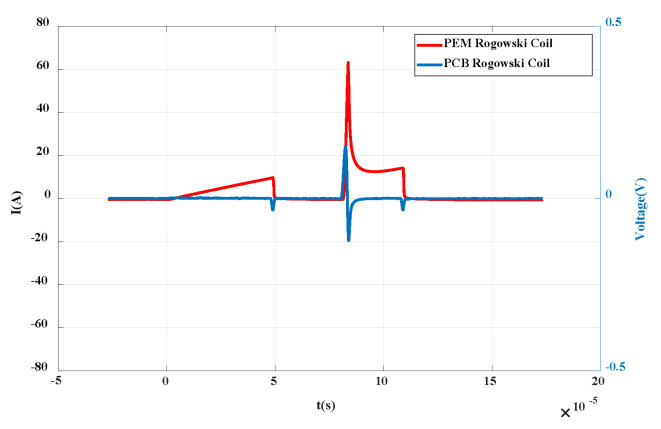
Differential current waveform of PPI.

**Figure 22 sensors-20-04080-f022:**
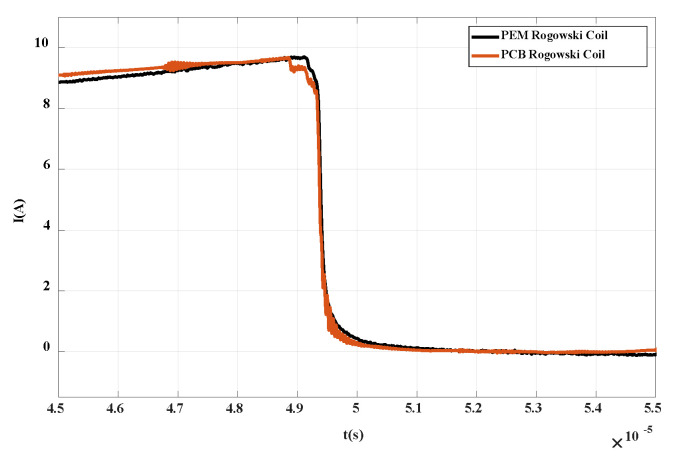
Turn-off current waveform of PPI.

**Figure 23 sensors-20-04080-f023:**
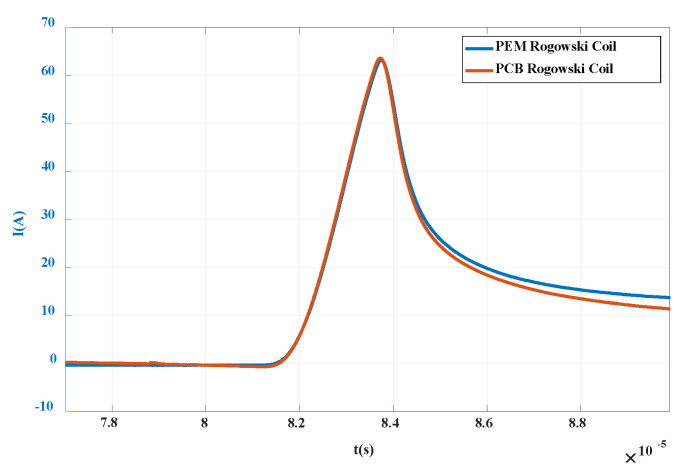
Turn-on current waveform of PPI.

**Table 1 sensors-20-04080-t001:** Simulation results of the mutual inductance (nH) between the 4 coils and 11 bosses.

Name	Coil1	Coil2	Coil3	Coil4
B1	0.0444	0.0287	0.0198	0.0347
B2	0.0383	0.0205	0.0013	0.0342
B3	0.0292	0.0191	0.0160	2.7923
B4	0.0337	0.0261	0.0349	0.0055
B5	0.0322	0.0258	2.8324	0.0159
B6	0.0466	0.0536	0.0220	0.0195
B7	0.0203	0.0689	0.0072	0.0111
B8	0.0161	2.7808	0.0450	0.0134
B9	2.7385	0.0046	0.0031	0.0104
B10	0.0428	0.0124	0.0050	0.0145
B11	0.0835	0.0076	0.0035	0.0145

**Table 2 sensors-20-04080-t002:** Parameter table of PCB Rogowski coil.

Turns	Inner Edge	Outer Edge	Thickness	Line Width	Internal Resistance	Capacitance	Self Inductance	Mutual Inductance
76	9.6 mm	15.8 mm	1 mm	3.5 mil	1.401 Ω	0.9 pF	411 nH	5.41 nH

**Table 3 sensors-20-04080-t003:** Main parameters of lightning surge generator.

Specification and Model	LSG-506B
the front	10 μs ± 30%
Pulse width	700 μs ± 20%
peak value	0–6 kV
Surge polarity	Positive/negative/positive negative alternation
Output impedance	40 Ω, 15 Ω ± 10%
atmospheric pressure	86–106 KPa
Rated working voltage	220 V ± 10% 50/60 Hz

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
