# Peer review of "Integrated Rogowski Coil Sensor for Press-Pack Insulated Gate Bipolar Transistor Chips"

_sensors, 2020, doi:10.3390/s20154080_

Round 1
Reviewer 1 Report
The research work and objectives are clear. It is also appropriate for the author to choose proposed method to solve the problem. However, there are still many problems to be improved.
1) The abstract should be rewritten, which only emphasizes the characteristics and difficulties of the research object, and the introduction of research methods is too simple.
2) It is suggested that the author give the design index, and the sensor design around the design index will be more targeted.
3) The analysis and discussion of the test results are not comprehensive enough. Especially, the author should go to the Metrology Department to calibrate the sensor before the Laboratory test.
Reviewer 2 Report
The originality/novelty of the paper cannot be claimed without comparing with the new research in the area.
The Introduction section should contain relevant recent research papers with new technology.
Some rearrangement of figures on the page is necessary (e.g. Fig.4). The figures with the simulation results should be increased.
Reviewer 3 Report
The topic of the manuscript “ Integrated Rogowski Coil Sensor for Press-Pack 2 Insulated Gate Bipolar Transistor Chips” is important and interesting. In my opinion, this study is a valuable work, the research is well designed, and decisions are justified. Still, the paper contains some weaknesses (I have listed below), I suggest this paper for “major revision”:
- The English of the paper is good but it contains a lot of miss typing! Please check it carefully!
- The Abstract is not well written. It contains much more information about the general topic of the paper than about the work of the authors. Please reduce a bit of the general content and give more information about your contribution.
- The working principle of the Rogowski sensors is presented nicely; however, it is too long. It such a basic knowledge that the readers can find it on the web. So you could reduce it.
- Please provide the units in Table 1.
- The organization of Table 2 is confusing, please fix it!
- 11 is too small, the captions are unreadable!
- 13-16 are too small too, and please change the sequence of the figure caption and the sub-figures captions!
- 18 is unnecessary, it does not contain any information for the readers. A Block diagram would be better!
- 23 contains some chinse writing, please correct it.
- 24, the same comment as at Fig. 18.
- The Conclusions are not well written. It is too short and too general! Please write a real conclusion according to you main findings! The Article is relatively long with a lot of findings… So use them!
Round 2
Reviewer 1 Report
The author has made targeted amendments according to the opinions, and there is no new revision opinions.
Reviewer 2 Report
The paper has been improved by the authors, according to the reviewer's remarks.
Reviewer 3 Report
The paper has been improved, it is acceptable for publication.